# Personal Digital Twin: A Close Look into the Present and a Step towards the Future of Personalised Healthcare Industry

**DOI:** 10.3390/s22155918

**Published:** 2022-08-08

**Authors:** Radhya Sahal, Saeed H. Alsamhi, Kenneth N. Brown

**Affiliations:** 1School of Computer Science and Information Technology, University College Cork, T12 E8YV Cork, Ireland; 2Insight Centre for Data Analytics, National University of Ireland, N37 W089 Galway, Ireland; 3Faculty of Engineering, IBB University, Ibb 70270, Yemen

**Keywords:** personalised healthcare, digital twin, personal digital twin, data analysis, COVID-19

## Abstract

Digital twins (DTs) play a vital role in revolutionising the healthcare industry, leading to more personalised, intelligent, and proactive healthcare. With the evolution of personalised healthcare, there is a significant need to represent a virtual replica for individuals to provide the right type of care in the right way and at the right time. Therefore, in this paper, we surveyed the concept of a personal digital twin (PDT) as an enhanced version of the DT with actionable insight capabilities. In particular, PDT can bring value to patients by enabling more accurate decision making and proper treatment selection and optimisation. Then, we explored the progression of PDT as a revolutionary technology in healthcare research and industry. However, although several research works have been performed for smart healthcare using DT, PDT is still at an early stage. Consequently, we believe that this work can be a step towards smart personalised healthcare industry by guiding the design of industrial personalised healthcare systems. Accordingly, we introduced a reference framework that empowers smart personalised healthcare using PDTs by bringing together existing advanced technologies (i.e., DT, blockchain, and AI). Then, we described some selected use cases, including the mitigation of COVID-19 contagion, COVID-19 survivor follow-up care, personalised COVID-19 medicine, personalised osteoporosis prevention, personalised cancer survivor follow-up care, and personalised nutrition. Finally, we identified further challenges to pave the PDT paradigm toward the smart personalised healthcare industry.

## 1. Introduction

The term “digital twin”, DT, was first coined by NASA for a virtual replica of a physical structure in real space [1]. Then, multiple definitions are used to define DTs for research and industry [2]. According to the literature research, we used the definition that defines DT as a digital replica of the physical thing (i.e., a physical twin, which could be a device, machine, person, etc.). Based on this definition, the DT has these features [3,4]; (1) DT can represent the physical twin in the real world; (2) the data-driven DT contains all processes and operations related to the physical twin; (3) the DT always has up-to-date data about its physical twin which means that the DT continuously synchronises with its physical twin; and (4) it shows the simulated physical twin behaviour (see Figure 1).

Most research and industry have recently adopted DT technology from a “disease care” perspective [5,6]. Moreover, they started to integrate DT technology with emerging technologies (e.g., Industrial Internet of Things (IIoT), blockchain, and artificial intelligence (AI)) for personalised healthcare purposes, which is person-centric prevention [7,8]. With the evolution of the personalised healthcare industry, everyone can be flanked by a DT, representing their healthcare markers. Furthermore, every person is unique in many way: for example, the symptoms of diseases slightly differ from one person to another person [9]. Therefore, most researchers and industrial bloggers have early brief definitions for PDT based on information across biological scales, from genotype to phenotype [10,11]. However, we believe that not only a set of biological data but also PDT could represent a person as a human by reflecting the different aspects of their life from birth throughout their lifetime. Substantially, the PDT will change our future life in terms of life, work, contact, love, fighting, learning, fun, play, and death. Consequently, we comprehensively define the PDT concept from different perspectives, including its mental, physical, social, and biological aspects. Figure 2 depicts our conceptual definition of PDT, including (1) mental activities (e.g., thinking, ideas, thoughts, and knowledge); (2) physical activities (e.g., sport, hobbies, dietary habits, sleeping patterns, etc.); (3) social activities including life interactions with people and virtual social networks; and (4) biological scales, i.e., representing vital organs such as heart, lungs, liver, brain, and the kidneys.

Without loss of generality, based on the biological perspective, a PDT for a patient’s heart could be built with heart-related medical measurements (e.g., heart rate, electrocardiogram results, cardiac MR results, blood pressure, and genetic information) [12]. The PDT will clearly show how the patient’s heart works. It will help cardiologists by showing the exact shape of the real patient’s heart as well as its dimensions, partitions, ejection fraction, electrical signal activation, and blood pressure variation. This will help cardiologists control the functioning of a patient’s heart. Furthermore, this will help improve and personalise the medicine and optimise the treatment planning before actual treatment delivery. Undoubtedly, a PDT for the heart is an incredibly individualised approach that can significantly contribute to cardiac health.

On the other hand, in manufacturing, the deployed DTs are used to assess the health conditions of machines and then predict the potential risks by estimating a particular time for the next needed maintenance [2,13]. In smart personalised healthcare industry, combining PDT data with the analytics technologies, business rules, and optimisation algorithms can support human decisions or even automate decision making regarding personalised and effective care. In particular, the collected data from the PDT are sent to large data pools to feed analytical engines and provide a deeper understanding of individual recommendation care. Consequently, better knowledge of the person’s health status using their PDT can help diagnose the potential health risks early, improve individualised clinical pathways, and support treatment planning, therapy, and lifestyle intervention. Furthermore, by combining PDT with blockchain technology, the PDT can maintain a high level of privacy and trust for handling personal patient data [14]. In particular, the PDT provides a secure personal digital replica of a specific patient created by their medical data. However, the PDT is still in early its early stages and several years away from completion. Furthermore, the improvement of personal patient care and healthy living by bridging the physical and digital worlds still faces a great many challenges.

### 1.1. Motivation

Our main motivation for conducting this research is to introduce a PDT concept to empower personalised healthcare for better patient care and healthy living. Furthermore, we believe that this work could be a step towards smart personalised healthcare industry by guiding the design of industrial personalised healthcare systems. Furthermore, we will start by digging deeper into the motivation behind introducing PDT as follows:A personal digital model for a person is a complicated task compared with that for an engine due to the differences among people. Therefore, the ability to distinguish individual persons could enable smart personalised healthcare to enhance and improve prevention in patients’ futures.No one-size-fits-all personalised healthcare procedure exists that includes personalised diagnosis, therapy selection, treatment planning guidance, nutrition, and mental well-being based on the patient’s physical characteristics, medical history, current condition, and future needs.There is no reference framework for the data-driven-based personalised healthcare industry. Most researchers adopt DTs for healthcare from a specific perspective (e.g., personalised medicine [9], specific chronic disease diagnosis (including heart disease, stroke, cancer, osteoporosis, etc.), and personalised nutrition [15]). There are no identified requirements to implement the PDT system for the industrial personalised healthcare system.

### 1.2. Contribution

The novelty of using DT technology in healthcare is evidenced by a large number of research works published in the last decade. Furthermore, DT technology has multilateral applications, which makes it a frontier technology for the healthcare industry. Therefore, without taking away from the novelty of the digital twin in the healthcare industry, it seems reasonable to realise that personalised solutions based on DTs are in huge demand. However, personalised solutions are partly already in place but are still in their infant stages. Consequently, this is why we believe that the PDT is a step towards a smart personalised healthcare industry by looking at its current progression and the future of the personalised healthcare industry. Furthermore, PDT-based personalised solutions need frameworks for designing and developing to work in smart personalised healthcare ecosystems.

To the best of our knowledge, no framework based on PDT has been proposed to design a smart personalised healthcare system. Furthermore, it is too early to have a baseline to identify how PDTs will be implemented. This is due to the highly diverse nature of healthcare delivery systems. This motivates us to propose a reference framework to integrate PDTs with patient data intelligence to provide a baseline for a smart personalised healthcare industry. The proposed framework aims to empower smart personalised healthcare for person-centric prevention and care, including rapid diagnosis, treatment, the early prevention of diseases and other personalised healthcare issues.

Our main contributions in this paper can be summarised as follows:We introduced the concept of PDT as an enhanced version of the DT which has personalised and actionable insights capabilities that improve personalised healthcare. Then, we provide its benefits for the smart personalised healthcare industry.We explore the progression of PDT as a revolutionary technology in healthcare research and industry.We propose a reference framework for smart personalised healthcare which aims to bring together existing advanced technologies (e.g., DT, AI, and blockchain). The proposed framework aims to improve personalised healthcare by supporting patient-centred care as a reality in everyday life, including physician–patient communication and facilitating shared decision making. Furthermore, we identify high-level functional requirements for building a smart personalised healthcare system.We provide some selected use cases of adopting PDTs in personalised healthcare, including the mitigation of COVID-19 contagion, COVID-19 survivor follow-up care, personalised COVID-19 medicine, personalised osteoporosis prevention, and personalised cancer survivor follow-up care and personalised nutrition.

### 1.3. Paper Organisation

The remainder of this paper is organised as shown in Figure 3. The current state of research is provided in Section 2. The research questions, objectives, and methodology are presented in Section 3. Then, the PDT concept and its benefits are introduced in Section 4. The progression of PDT in healthcare research and industry is provided in Section 5. The proposed reference framework for smart personalised healthcare industry is described in Section 6. The focus on personalised healthcare use cases is presented in Section 7. The validation and open challenges are discussed in Section 8. Finally, conclusions are presented in Section 9.

## 2. State of Research

DTs are used in healthcare for building a digital representation of healthcare data, including electronic medical data, hospital environments, human physiology, operational staff, and lab results [16]. Furthermore, it is used to create a virtual replica of a healthcare system which helps healthcare organisations review operational strategies, utilise capacities, and evaluate staffing performance. On the other hand, at the individual level, which is our focus in this work, DTs are applied for personalised diagnosis, treatment planning, personalised care, etc. Accordingly, we will discuss a few existing studies to showcase how DT technology is applied in personalised healthcare.

For adopting DTs for general-purpose use in healthcare, Angulo et al. [16] introduced a general-purpose proposal for creating DTs which apply to the health field, specifically lung cancer patients. Furthermore, Shengli [17] introduced a conceptual model and the characteristics of the human digital twin (HDT). Furthermore, the authors in [18] introduced the concept of a well-being digital twin (WDT), its architecture and impacts.

In the context of personalised medical treatment purposes, the authors in [19] have provided a narrative review about existing and future opportunities to capture clinical digital biomarkers in the care of people with multiple sclerosis disease. Furthermore, they have introduced DTs for multiple sclerosis (DTMS) to monitor the long-term multiple sclerosis disease. DTMS is used for personalised treatment by collecting high-frequency and structured patient data to propose a tailored therapy. Furthermore, the authors in [20] introduced the DT concept for personalised medicine. Moreover, they addressed the expanding DTs by integrating variables of multiple types, locations, and time points. Moreover, the authors introduced the steps of the Swedish Digital Twin Consortium (SDTC) strategy in which (i) a DT is created for individual patients which contains unlimited copies based on the computational network models of thousands of disease-relevant variables; (ii) each twin is computationally treated with thousands of drugs to find the optimal drug for this patient; and (iii) the best drug that has the best effects is selected for this patient. Furthermore, Rivera et al. [21] presented their vision for applying the DT concept in personalised medical treatment. Finally, the authors elaborated on the definition of internal structures for DT to support precision medicine techniques. Considering the decision support purpose, the authors in [22] have introduced a patient-specific finite element model approach based on DTs to help personalised clinical decision making. This approach aims to optimise trauma surgery and postoperative management by focusing on tibial plateau fractures to enhance biomechanical knowledge. It aims to optimise surgical trauma procedures and improve postoperative management decision making. Furthermore, the authors in [23] proposed a DT-based approach to improve healthcare decision support systems. These authors used state-of-the-art explainability concepts to interpret machine learning models to give doctors a more generic perspective that helps the diagnosis.

In the context of individual risk management, Ogunseiju et al. [24] proposed a DT-based framework to improve the self-management of ergonomic risks for construction work. The framework has been proposed for dynamic mapping between construction workers and their virtual replicas to assess the risks by monitoring the workers’ movement for musculoskeletal injury prevention. The authors demonstrated the feasibility of their proposed approach by evaluating the LSTM deep learning technique on participants’ movement data captured using wearable devices. Furthermore, the authors in [25] presented their vision of agent-based DT in the healthcare context. They described a case study of agent-based DT for supporting the process of severe trauma management.

In adopting DTs to specifically protect against COVID-19 in infected patients, Laubenbacher et al. [10] discussed how medical DTs are beneficial to mitigate COVID-19 viral infection and any future pandemic. Furthermore, they were concerned with combining medical DTs with a mechanistic understanding of the physiology and viral replication and AI techniques for optimising the treatment of patients infected with a virus. Furthermore, in our previous work, we introduced a blockchain-based collaborative DTs framework for decentralised epidemic alerting to protect against COVID-19 and any future pandemics [26]. The proposed framework utilises the data-driven digital twins collaboration with the help of blockchain technology to protect against pandemics such as COVID-19.

Because only a few publications exist on the adoption of PDTs in healthcare, PDTs for personalised healthcare have not yet been a big focus in the literature to date. Furthermore, personal data-by-design concepts have not been considered yet. Moreover, comparing the previous related works to our work, we introduced the concept of PDT, an enhanced version of the DTs, and its benefits for smart personalised healthcare. Furthermore, we proposed a comprehensive reference framework for smart personalised healthcare, which aims to bring together existing advanced technologies, including DTs, blockchain, and AI. Furthermore, we provided some selected use cases of adopting PDTs in personalised healthcare.

### Summary

In this section, we explore several research works regarding the adoption of DT technology in the personalised healthcare industry. In brief, Table 1 shows the contributions of the previous research works in personalised healthcare and their limitations. Furthermore, Table 2 describes a comparison of the current work and the present work concerning the DTs applications, data analysis (i.e., AI, XAI), applications, and use cases.

## 3. Research Questions, Objectives and Methodology

In this section, we state the research questions, objectives and methodology that we worked with in this research paper.

### 3.1. Research Questions

The PDT represents the diverse and real-time information about a person obtained by wearable devices. Utilising PDT data can improve the smart personalised healthcare industry. Therefore, healthcare providers are eager to have personalised information about their clients (i.e., patients) to make decisions and recommendations. The research based on PDT is still in its early stage of exploring the opportunities of adopting PDT technologies for smart personalised healthcare. Accordingly, this paper aimed to overview the PDT concept and its benefits for healthcare, revolutionise the healthcare industry, the reference framework and the requirements to build a PDT-based system for smart personalised healthcare, the potential applications and use cases, and the open challenges for adopting PDT in the healthcare industry. This is achieved by investigating the following research questions, which are summarised in Table 3.
**RQ1: What are the role and benefits of introducing PDT?**The purpose of this question aims to provide an overview of the PDT’s role and its benefits in healthcare. To address this question, we introduce an overview of the PDT’s role and we explore its benefits for healthcare (see Section 4).**RQ2: How could PDT revolutionise the personalised healthcare industry?**This question aims to explore the PDT industry’s progress concerning smart personalised healthcare. To address this question, we search for the healthcare companies and the ongoing research projects and centres that adopt digital twins and collaborate with industry partners (see Section 5).**RQ3: What are the requirements for building a PDT-based system for a smart personalised healthcare industry?** The purpose of this question aims to identify the requirements for building a PDT system for smart personalised healthcare. To address this question, we identified a set of high-level requirements to fulfil the criteria of building a PDT-based smart personalised healthcare system (see Section 6.1).**RQ4: What are the key layers for implementing a PDT-based smart personalised healthcare system?** The purpose of this question aims to identify the key modules/layers for building a PDT system for smart personalised healthcare. To address this question, we propose a reference framework to introduce the key modules/layers to implement a PDT as an enhanced version of the digital twins that has personalised and actionable insight capabilities for improving personalised healthcare (see Section 6.2).**RQ5: What are the potential applications of using PDT for a smart personalised healthcare industry?** The purpose of this question aims to identify the potential applications of PDT being used for smart personalised healthcare. To address this question, we discuss the potential applications of using PDT, such as personalised diagnosis treatment for the early prevention of diseases (see Section 6.2.3).**RQ6: How is the PDT concept being applied to protect against the COVID-19 outbreak and any future pandemic?** The purpose of this question is to elaborate on how PDT capabilities can be used to protect against COVID-19 and any future pandemic. To address this question, we describe how the proposed reference framework can be applied to help mitigate COVID-19 contagion (see Section 7.1).**RQ7: What are the open challenges to applying PDT in smart personalised healthcare?** The purpose of this question is to explore the open challenges of using smart personalised healthcare.

To address these questions, we discuss the challenges of applying PDT for smart personalised healthcare in terms of data privacy and regulations, data quality, ethics issues, modelling, connectivity, timing, speed, and technical issues (see Section 8.2).

### 3.2. Research Objectives

We discuss the PDT concept and introduce the benefits of using PDT to answer these questions. Then, we proposed a reference framework for PDT being used for smart personalised healthcare. The inputs will be the collected healthcare data from the medical sensors and wearable devices such as temperature, heart pulse, blood pressure, insulin, and other fitness activities. The outputs will be insights based on a data-driven PDT that could be used for any healthcare application such as smart personalised healthcare systems and smart hospitals. Furthermore, these insights could be used in different smart personalised healthcare applications and to automate decision making within personalised healthcare systems. Furthermore, we selected some use cases concerning personalised healthcare. Moreover, based on the result of the COVID-19 pandemic outbreak, there is a significant need to adopt PDT to protect against this pandemic crisis. Therefore, we also describe how the proposed reference framework can be applied to mitigate COVID-19 contagion and help patients with long-haul COVID-19 while preserving personalised healthcare.

### 3.3. Research Methodology

Figure 4 shows the key steps in conducting this study. First, a literature review on the state of the art of DTs adoption in personalised healthcare is provided. Then, a comprehensive definition of the PDT concept is introduced to identify its role and benefits for the personalised healthcare industry. Furthermore, a brief overview of PDT progression is provided to understand how the concept has revolutionised personalised healthcare, including research organisations and the industry sector. Then, a set of high-level requirements for smart personalised healthcare is introduced, followed by a detailed description of the key modules/layers used to elaborate the proposed framework. Afterwards, the potential personalised healthcare use cases are demonstrated.

#### Summary

In this section, we identify the research questions that are answered in the following sections (see Table 3). Then, we discuss the research objectives by concisely describing what the research is trying to answer with regard to the PDT concept, its reference framework, as well as its potential applications and use cases. Furthermore, we highlight the methodology—including the critical steps in conducting this study—to analyse information about the PDT topic (see Figure 4).

## 4. Personal Digital Twin Concept and Its Benefits


**RQ1: What are the role and benefits of introducing PDT?**


This section introduces an overview of the PDT role in healthcare and the high-level view of PDT. Then, we provide a set of positive aspects of using PDT in healthcare.

### 4.1. The Role of Personal Digital Twin

Substantially, people are already digital replicas on their social media networks. These social digital replicas show people’s thinking, opinions, feelings, activities, etc. Similarly, PDT represents a virtual replica of a human [17] including their mental, biological, physical, and social aspects. Therefore, the role of PDT is to make a digital version of a human life to help them with self-care, self-reflection, and personal growth. Furthermore, the PDTs present people’s current lives and can provide life forever through their data. On the bright side of PDT, PDT is gaining momentum in the healthcare industry. It revolutionises personalised healthcare by delivering live, personal health data sources for healthcare-based learning systems to predict the potential risks, especially for older people and chronic diseases. This attracts healthcare providers and stakeholders to maximise their business by making personalised decisions and recommendations for their clients. Beyond this, PDT can help people improve their personal productivity development and increase their health longevity by following efficient self-care routines in their daily life. Unfortunately, PDT has critical privacy issues. The ethical conditions related to accessing PDTs should be considered when PDTs share people’s sensitive and personal content.

The high level of PDT is shown in Figure 5. It can be seen that the high level of PDT contains the physical world and cyber world. The data are captured from physical smartphones and wearable devices such as smartwatches and Fitbit for the physical world. In particular, these personal data are collected from physical sensors that record temperature, heart pulse, blood pressure, blood sugar levels, insulin, number of walking steps, and other activities.

### 4.2. The Benefits of Introducing Personal Digital Twin

There are a set of benefits of using PDT, including building a digital patient model, personalised treatment, rapid diagnosis, predicting responses to surgical interventions, joint research, empowering the world of AI-enhanced humanity, empowering self-reflection and self-coaching, and human immortality through data (see Figure 6). The details regarding these benefits of using PDT are listed as follows.
**Building digital patient model:** The digital patient integrates the different measurements of a person over time. PDT can help build a digital model to provide a big picture of a patient. This allows for bringing together all the information about a particular patient. This helps general practitioners to use a model or sub-model of a patient or a patient’s body part, such as an organ, and how it works over time. For example, dynamically updated digital body parts (e.g., a digital heart model, digital brain model, and digital liver) could support the early diagnosis and treatment planning for chronic diseases [12]. Furthermore, the digital patient model based on PDT would help predict which patients will fall ill weeks or months in advance. Moreover, the healthcare providers can access patients’ PDTs which contain personalised information about their health conditions to make appropriate decisions and personalised recommendations.**Personalised treatment:** As every person is unique, their immune system reacts to different diseases and differs from other people. Therefore, using PDTs to collect personal healthcare data about patients and then analyse them with AI techniques will provide insightful information about patients’ health conditions. This attracts the healthcare providers and pharma companies to utilise the PDT-based health data for individually prescribing drugs, e.g., unique drugs for each patient, and recommending an optimal therapy and improving the care for every customer. Furthermore, the individual treatment based on PDT would help predict how a particular patient will react to a specific treatment, how they can most benefit, and what the side effects are. These predicted remarks could even further revolutionise medicine and maximise the profits of healthcare enterprises and pharma companies.**Rapid diagnosis:** PDTs could be used to diagnose the potential risks for chronic diseases by analysing the patients’ data. For example, PDT-based machine learning models are used to understand patterns and use predictions to help in the early diagnosis of cancer, asthma, diabetes, heart disease, multiple sclerosis, etc. [12,19].**Predicting responses to surgical interventions:** PDTs could be used to simulate the individual procedures of surgery. The individual-simulated surgery based on PDTs considering the personalised circumstances of a particular patient helps avoid the potential risks and identify the optimal devices and techniques for surgical procedures.**Joint research opportunity:** Based on the genetic background and medical history, researchers can perform their experimental work, including individualised treatment simulations using PDTs to determine the best therapy option for individual patients. The PDT provides researchers with a whole image of the human body, which gives a set of relationships between human organs and the interactions with different diseases, nutrition, and lifestyle. These relationships can offer opportunities for joint research on a larger scale by clinicians, scientists, engineers, and healthcare technology providers. For example, there is a clinical relationship between knee osteoarthritis, cardiovascular diseases, and sleep disorders, which can offer elaborate translational research possibilities for knee therapy.**Empowering the world of AI-enhanced humanity:** PDTs can mirror humans’ body parts, organs, and personal genomes. However, on the other hand, AI plays a vital role in contributing to human healthcare by utilising the massive amounts of data that their PDTs may capture. Consequently, healthcare leaders are taking advantage of data from PDTs and then applying AI to build a big picture about individual clients and their personhoods to deliver enhanced care services.**Empowering self-reflection and self-coaching:** PDTs can contribute to human mental health by contacting people such as personal coaches, leadership trainers, and behavioural therapists to realise their weaknesses and strengths. For example, Mind Bank Ai (https://www.mindbank.ai/mental-health.html, accessed on 15 February 2022) has designed PDT to individually assist people by giving them loop feedback about themselves gaining mental strength through self-discovery.**Human immortality through data:** We saw that social media can store people’s posts, including their voices, pictures, stories, opinions, thinking, and feeling. These data could be stored in their PDTs to maintain the footprint of their existence. Therefore, the PDTs will be the data lake for human eternity through data. The PDTs will provide a rich source of personal data that represent people. Furthermore, combining PDTs with NLP technology will be an interesting research direction for storytelling about people’s current lives, even forever on their behaviour.

#### Summary

In this section, we introduce the concept of the PDT role in personalised healthcare. Then, we provide a detailed description of the beneficial use of PDT. Finally, the info-graphic of the PDT benefits is depicted in Figure 6.

## 5. The Progression of Personal Digital Twin in Healthcare Research and Industry


**RQ2: How could PDT revolutionise the personalised healthcare industry?**


This section provides an overview of the PDT industry’s progress in smart personalised healthcare. First, ongoing research projects and centres collaborating with industry partners are provided. Then, a summary of healthcare companies that have adopted DT is introduced and categorised based on their products and services.

### 5.1. Healthcare Research Centres and Projects

Some research centres and projects adopted DT technology to improve their studies, including in personalised diagnosis, treatment, and medicine. Some examples include the Swedish Digital Twin Consortium (SDTC), Human Digital Twin OnePlanet research centre, Empa research centre, DIGIPREDICT consortium, Living Heart project, and COVID-19 Long-hauler project. Further details about these research centres and projects are provided below. Furthermore, a summary of the DT-based healthcare research centres and projects and their focus is depicted in Figure 7.
In **Swedish Digital Twin Consortium** (https://liu.se/en/news-item/digital-tvillingar-hjalpmedel-for-skraddarsydd-medicinering-, accessed on 1 February 2022), the Swedish researchers adopted DT technology for personalised medicine using RNA. The SDTC (https://www.sdtc.se/, accessed on 1 February 2022) aims to develop a strategy for personalised medicine [20]. The SDTC strategy is based on three steps: (i) creating a DT for individual patients which contains unlimited copies based on the computational network models of thousands of disease-relevant variables; (ii) each twin being computationally treated with thousands of drugs to find the optimal drug for this patient; and (iii) the best drug which has the best effects is selected for this patient.**Human Digital Twin, OnePlanet Research Center** (https://oneplanetresearch.nl/innovatie/digital-twin/, accessed on 2 February 2022) developed an AI-guided digital platform for continuous collection and the analysis of health and nutrition data using sensors. The digital data platform is being constructed using health data collected in OnePlanet’s innovation programs Ingestibles for Gut Health, Smart Bathroom for Health and Studies in Nutrition & Mental Wellbeing. It is a collaborative research work between digital platform experts from imec, specialising in high-tech sensors and wearables, nutritionists, behavioural experts and doctors from Wageningen University & Research, Radboud University and Radboudu. The DT technology serves the research platform in this research centre by collecting health and nutrition data to facilitate the early detection of diseases (e.g., diabetes, cardiovascular diseases, and burnout) and develop personalised products and services.**Empa research centre** (https://www.empa.ch/web/s604/eq71-digital-twin, accessed on 2 February 2022) in Switzerland utilises DT capabilities to improve the dosage of drugs for people afflicted by chronic pain. They studied some characteristics such as age and lifestyle to help them customise the DTs of patients and then predict the effects of pain medications. Then, the patients can report the effectiveness of their personalised dosages, which improves their DTs’ accuracy.**DIGIPREDICT consortium** (https://www.digipredict.eu/, accessed on 3 February 2022) is a research project with seven top-level universities, research centres, hospitals, and three SMEs. The DIGIPREDICT partners are working to combine cross-cutting lines of biomedical research by bringing a range of excellent international scientists with complementary and interdisciplinary skills. The DIGIPREDICT proposes the first DT of its kind that predicts the progression of the disease and the need for early intervention in infectious and cardiovascular diseases. With regard to the development work, the DIGIPREDICT DT started to predict whether COVID-19 patients will develop severe cardiovascular complications and, in the long term, the possibility of the onset of inflammatory disease.**Living Heart project** (https://www.3ds.com/products-services/simulia/solutions/life-sciences-healthcare/the-living-heart-project/, accessed on 3 February 2022) was launched by Dassault Systèmes in 2014. The project aims to obtain information about the human heart using its virtual image, i.e., digital heart twin. The project is an open source collaboration between medical researchers and industry partners, including surgeons, medical device manufacturers, and drug companies.**COVID-19 Long-hauler project** (https://www.delltechnologies.com/asset/en-us/solutions/business-solutions/briefs-summaries/dell-i2b2-infographic.pdf, accessed on 3 February 2022) is a collaborative research work between Dell Technologies and i2b2 tranSMART. The project aims to apply AI with advanced technology such as DTs to understand the causes of the post-acute sequelae of SARS-CoV-2 (PASC) and develop effective treatments. The DTs will be shared with researchers from more than 200 hospitals and research centres. The DTs allow the researchers to conduct millions of simulations to identify the best treatments for COVID-19 long-haulers.

### 5.2. Healthcare Industry

The healthcare industry is adopting emerging technologies such as IoT, AI, and DTs to improve business. For example, they utilise the virtual replicas of their clients (i.e., provided by DT technologies) to monitor their health status and provide care services based on individual needs. In particular, the healthcare industry uses people’s personal information extracted from their PDTs. Then, they analyse the PDT-based high-quality collected health data about the individual patient (e.g., biometrical, cognitive and genetic) to predict the potential risks and track progress over time. These tracked progression and the predictions are considered gold indicators for (i) people who want to practise self-care; (ii) healthcare insurance companies; and (iii) health practitioners who prefer to follow their patients and have insightful information about their health history.

We listed the progression of some healthcare and pharma companies using DT technology for smart personalised healthcare (see Table 4). Some companies target a specific vital part such as the heart or brain, while others provide a generic product to improve personalised healthcare. Based on the type of products and services, FEops [27], Simens [11], Philips [28], and Dassault Systèmes [29] are using DT technology for heart virtualisation to empower personalised treatment for patients. For example, both Simens and Dassault Systèmes provide a 3D model for a live heart for cardiac treatment and research purposes. On the other hand, Living Brain [30] provides a tracking progression of neurodegenerative disease. Moreover, IBM [31], Babylon [32] and DigiTwin [33] use DT technology to deliver personalised healthcare services to their clients.

#### Summary

This section provides a comprehensive study of the progression of PDT in research and industry. The infographic about the set of ongoing research projects and centres is depicted in Figure 7. Furthermore, we provide a summary of the healthcare companies that adopted DT in terms of their products and services (see Table 4).

## 6. Proposed Reference Framework for Smart Personalised Healthcare Industry

The merit of the proposed framework is integrating DT, AI, blockchain technologies, and the operational data of patients to deliver smart personalised healthcare services to improve people’s lives. Therefore, the proposed framework is considered one level higher than the adoption of DT technology in the healthcare industry. Furthermore, the proposed framework could be developed and implemented on top of DT platforms, which exploit AI capabilities to deliver smart personalised healthcare services. Moreover, the proposed framework was introduced by identifying the high-level requirements for smart personalised healthcare and the layers used to elaborate the framework.

### 6.1. High-Level Requirements for Smart Personalised Healthcare


**RQ3: What are the requirements for building a PDT-based system for a smart personalised healthcare industry?**


We identified the requirements for building a smart personalised healthcare system based on PDTs. Table 5 summarises 12 criteria to fulfil the high-level requirements for smart personalised healthcare, including data collection (R1), data update frequency (R2), data management (R3), data analysis (R4), data explainability (R5), data quality (R6), simulation capabilities (R7), privacy and confidentiality (R8), authorisation (R9), connectivity (R10), decision making (R11), and computing paradigm (R12).

The data are collected by the physical devices (e.g., on-body medical sensors, wearable devices, and personal activities) (R1). The collected data are of two types: historical data and real-time data. The historical data are collected from the medical records stored in the patients’ databases and medical ledgers. The real-time data are captured from remote sensors such as attached medical, wearable, and smartphones which publish their data to PDTs. Furthermore, the collected data could be managed and queried by real-time query engines and modelled based on the required specifications of the personalised healthcare system (R3). For example, if the smart personalised system is used for clinical diagnosis, the schema of PDT will be defined based on the required information for diagnosis. Another example is that, if the smart personalised system is used for fitness recommendations, the schema of PDT will be defined based on the necessary information about the physical activities captured by smartphones and wearable devices.

Based on the dynamism of the personal data, the data are frequently updated in real time, which is beneficial to the healthcare timely decision-making process (R2 and R11). The PDTs provide the decision-making participants (e.g., the patients, doctors, hospital, clinical research systems, diagnostic laboratories, and healthcare providers) with insightful information for making decisions, including the healthcare recommendations, rapid diagnosis, treatment plans, and potential diseases. These actionable decisions are made based on the predicted indicators obtained by data analysis technologies (i.e., machine learning (ML) and deep learning (DL)) trained by historical data and evaluated by continuous updating by the PDTs’ real-time data readings (R4). Moreover, specific analytical tasks need further explainability for some personalised healthcare use cases to support diagnosis and clinical decision system (R5) [34,35]. For example, clinical specialists need a data interpretation for the clinical significance of the individual patient.

Furthermore, they would like to individually conclude the diagnosis by reviewing the related cases for other patients and considering personalised biophysical models and health conditions for the individual patient. Furthermore, poor quality data can lead to poor treatment of the patient, e.g., inaccurate diagnosis and improper recommendation (R6) [36]. Therefore, the data should be as clean and free of errors as possible to allow the doctors to make more effective and informed decisions for their patients and enable healthcare providers to deliver highly personalised healthcare services to their clients.

However, simulation aims to investigate what the patient has, but it could also help understand what could happen in the future (R7). For example, using PDTs’ capabilities with medical measurements (e.g., heart rate, blood pressure, and insulin level) could simulate some clinical scenarios to select the optimal diagnosis. Another example is using individual simulated surgery to predict the responses to surgical interventions; the doctors will be ready for the appropriate actions during surgery.

Privacy and confidentiality are the most critical aspects of any healthcare system because they maintain their personal information, including medical records. The personal medical records should only be accessible by those legally authorised to access patients’ personal information (R8 and R9). Furthermore, the patients should explicitly consent to their data being shared with non-profit organisations and trusted research partners for research purposes. Furthermore, all regulatory bodies that govern the use of patients’ personal information must comply with the confidential agreements between parties.

The PDTs could not achieve their objectives in real time (R10) without good connectivity. Thus, the good connectivity of the physical devices (e.g., on-body and wearable sensors) is essential to avoid critical data loss, especially for health risk conditions. In addition, the PDT-based predictive data analysis could be performed on computing paradigms such as cloud and edge computing to leverage extra computing capabilities for real-time analysis (R12).

### 6.2. Layers of the Proposed Reference Framework


**RQ4: What are the key layers for implementing a PDT-based smart personalised healthcare system?**


This paper proposes a reference framework to build a smart personalised healthcare system based on PDT. The proposed framework aims to enable the healthcare industry to deliver better-personalised services to its clients. The proposed framework empowers grants the PDT more intelligence to provide a baseline for designing smart personalised healthcare systems. Three layers are used to equip the reference framework of building a PDT-based personalised healthcare system with operational data intelligence. As shown in Figure 8, the layers are (i) the physical devices containing the smartphones, wearable devices, medical sensors, etc.; (ii) industrial technologies; and (iii) application areas. These layers will be elaborated upon as follows.

#### 6.2.1. Physical Devices

Physical devices are potentially involved in the smart personalised healthcare system. The devices could be biosensors, medical devices, smartphones, wearable devices, etc. Biosensors are portable medical-grade wireless devices that discreetly adhere to the body. For example, the Philips Biosensor BX100 monitors the chest by measuring vital signs, posture, and activity while allowing patient mobility. The smartphones acts as an instrumental interface to mirror these biosensors to help aid the early identification of patient deterioration and then drive early intervention [37]. Furthermore, the built-in smartphone sensors (e.g., GPS, step counter, proximity and mobile health sensors) allow people to share their activities with the corresponding parties (e.g., doctors, hospitals, and healthcare providers.) for health-tracking purposes. Furthermore, wearable devices such as smartwatches and Fitbit devices could contribute by publishing personalised wearable data about people. For example, the real-time self-measurement of temperature, heart rate, blood pressure, glucose level, and sleep could be a point of care for patients to predict the potential risks and take early prevention actions.

#### 6.2.2. Industrial Technologies

Different industrial technologies are used for building a concert PDT model for personalise healthcare applications [9]. Consequently, this layer briefly highlights the role of emerging industrial technologies in building a smart personalised healthcare system. We describe this layer using eight components for simplification: collaborating twins, data management, data analysis, synchronisation, simulation, stream processing, blockchain technologies, and computing technologies.


**Collaborating twins**


Collaborating twins is an emerging area of interest among DTs in different sectors such as the energy industry—fault diagnosis of wind turbines [4]; the railway industry—predictive maintenance [38]; and the logistics industry [26]. As people collaborate to innovate and perform jobs, DTs do the same by sharing and exchanging information among entities and sharing tasks to act accordingly. An example of DTs collaboration is DTs for different aircraft components (e.g., wing, engine, and fuselage). First, the DTs of each machine collaborate to deliver insights into the state of the engine. Then, the data-driven DTs of the aircraft components are used to analyse purposes to reduce engine downtime, improve maintenance staff efficiency, and optimise the spare part inventory.

Consequently, collaboration is essential for a group of DTs to effectively and efficiently perform analysis, while a single cannot do [39]. Thus, PDTs are built on collaborating twins for biosensors, medical devices, smartphones, wearable devices, etc. These sets of DTs of different body parts help understand the whole body’s PDT status within personalised healthcare systems.


**Data management**


Data management has three sub-components: data acquisition, query, and modelling. The data acquisition technologies are used to gather the sensors’ data in the real world and inject them into the parallel simulated environment. Then, the collected data are used for real-time data analysis. The data query component actively retrieves information from the running database. The running database holds the values of the deployed DTs within the virtual cyber world, whether historical DTs stored in ledgers or streaming DTs are used to retrieve the up-to-date data [40]. The data modelling technologies are used to semantically model the PDT data by representing the features of personal health data, which are used for digital objects for a person [12,21]. The features of the collaborating twins and their relations determine the PDT model’s complexity and data structure choice.


**Data analysis**


Pairing AI with DT technologies creates new efficiencies for smart healthcare industries. For example, applying predictive data analytic techniques (e.g., ML and DL) using data-driven DTs provides predictions of the disease progression in near real time [41,42]. These predictions are used as biomarkers to support early clinical decision making and propose patient-specific therapy using existing drugs. The data analysis component has three sub-components: the predictive model, explainable AI, and knowledgebase.

**Predictive model** The designed PDT-based predictive models are used to understand patterns and predictions to help personalised healthcare services. Furthermore, building predictive models based on DT-based multiple scales data helps build concert computational representations of biological processes and body systems. These models could be customised per-patient models by integrating with DT-based personalised clinical data from individual patients. Then, the PDT model can be used to derive personalised predictions about diagnosis, prognosis, and the efficient optimisation of therapeutic interventions.

Figure 9 shows the workflow description of building a predictive model using data-driven PDTs. The workflow mainly consists of two phases: building an offline predictive model and deploying an online predictive model [43]. For the offline predictive model, we will use the ML/DL techniques (e.g., classifier). An offline predictive model will be developed and trained using personal historical data collected from PDT-based medical records. For the online predictive model, the developed predictive model could be evaluated at the edge level and then be used to predict the potential risks online using PDT-based real-time streaming data.

**Explainable AI** Explainable AI is a set of tools and frameworks to help data scientists understand and interpret ML models’ predictions. According to a personalised healthcare context, the explainability of models is needed to provide accurate recommendations and evidence for individual patients. Clinicians apply the diagnostic inference by extracting the association rules among multiple data types such as clinical records, sensor readings, social activity, and environmental factors. Therefore, PDT technology could gain importance in helping clinicians expand their diagnosis by providing real-time data feeds about the patients’ current conditions. Furthermore, combing PDTs with explainable AI could enhance diagnosis and personalised treatments by obtaining an accurate picture of the patient health and recommending the proper care at the right time [16,23,35].

**Knowledgebase** This component manages the knowledgebase within the proposed reference framework. According to the building of PDTs using a set of collaborating twins, each DT relation to other DTs is defined and stored in the knowledgebase, such as instance–instance relations, inheritance, parent–child relations, and the whole system to remain consistent. Then, the knowledgebase contains the set of knowledge rules and ontologies learned through relevant ML techniques from DT-based historical data [44]. Furthermore, knowledge rules are used to refine the knowledgebase by identifying new conditions using rule-based analysis.


**Synchronisation**


Healthcare professionals and providers deal with huge amounts of data such as clinical data lakes, EMRs, medications, and infinite health records of patients’ current statuses [45]. Furthermore, proper integration and synchronisation can help handle healthcare data. Therefore, adopting PDTs can play a vital role in removing frustration by always being in sync with the existing patients’ records. Furthermore, healthcare providers should not hesitate to step into the future by adopting PDT-based solutions to integrate and synchronise their data for greater consistency and better real-time data collection. This motivated us to propose a synchronisation component that should be implemented as a PDT-based synchronisation interface to keep a similar synchronisation consistency within a personalised healthcare system.


**Simulation**


With a simulation, doctors/engineers can test scenarios and conduct assessments on a simulated version of medical devices. For example, the cardiologist could simulate procedures using heart catheters or surgery for an individual patient to avoid risks and carry out the intervention as well as possible. According to the context of PDTs, collaborating twins receive real-time updates from the medical devices to simulate an upcoming, more accurate, and valuable intervention. Therefore, this component simulates the data exchange among collaborating twins and the entire personalised healthcare system.


**Streaming processing**


The merit of DTs relies on continuous data updates by ingesting stream data from sensors [46]. Therefore, the streaming data analysis adopts DT technology for the grantee to receive real-time data to analyse the status of the physical assets. Furthermore, combining the DT technology with streaming analytics offers several benefits in different real-world applications. In particular, the DT technology helps engineers implement a stateful model of the physical data sources that generates event streams while maintaining separate state information for each data source [47]. According to the context of PDTs, the stream processing component is responsible for ingesting streaming twin data for implementing the smart personalised healthcare system.


**Blockchain technology**


In several industry areas, DT technology has been linked with blockchain technology to connect multiple DTs using distributed ledger technology (DLT) [48]. For instance, we envision two possibilities for blockchain and DTs. First, the required personal data surrounding PDTs must be immutable (i.e., they cannot be modified), and the blockchain plays a vital role in securing them. Thus, instead of keeping it in a traditional database, one method saves PDT-based information in the blockchain. Second, the importance of combining PDT and blockchain technology is the need for the PDT to interact with doctors, hospitals, healthcare providers, etc. [26].

Furthermore, the PDT-based blockchain offers secure distributed operational data management and analytics across multiple participants [38,49]. Furthermore, blockchain technology enhances healthcare industries to provide adequate patient care and high-quality health facilities [50]. It can keep an incorruptible, decentralised, and transparent log of all patient data which places the patient at the centre of the healthcare ecosystem. Accordingly, Table 6 lists some healthcare/medical care companies which have adopted blockchain practices in their products and services (e.g., Akiri [51], BurstIQ [52], Factom [53], MEDICALCHAIN [54], Guardtime [55], Professional Credentials Exchange [56], Avaneer Health [57], Coral Health [58], Robomed [59], and Patientory [60]).

According to the context of this work, Figure 10 depicts the participants collaborating in the smart personalised healthcare system. These participants are PDTs composed of a set of collaborating twins, operational staff (e.g., doctors, pharmacists, lab technicians, and scientists), public healthcare authorities (e.g., hospital and health organisations), and the healthcare industry. In particular, the decentralised nature of blockchain technology allows patients, doctors, and healthcare providers to share the same information quickly and safely by exchanging their data using a blockchain network. Furthermore, blockchain technology allows developers to implement smart contracts between PDT-based participants such as patients, i.e., PDT, and healthcare professionals to ensure data and treatments.


**Computing technology**


Due to the generation of large volumes of healthcare industry data, the DT-based predictive data analysis could be performed on computing paradigms such as cloud and edge computing to leverage extra computing capabilities for real-time analysis [61,62].

#### 6.2.3. Application Areas


**RQ5: What are the potential applications of using PDT for a smart personalised healthcare industry?**


PDT can be used in many industrial healthcare applications (e.g., personalised healthcare, smart hospital, e-healthcare, and decision making). In particular, the proposed reference framework could be applied to different areas because it can collect, manage, and analyse vast amount of personal healthcare data. For example, to facilitate healthcare-based decision-making applications, the PDT-based predictive analysis of healthcare data is applied to early-stage disease risk prediction, treatment planning and patient-specific recommendations. Moreover, based on the analysis of PDT discussed in this paper thus far, we summarise the potential smart personalised healthcare applications: personalised medicine, rapid diagnosis, self-care, remote care, fitness tracker, well-being, medical alerts, and pandemic combating (see Figure 11).
**Personalised medicine applications:** Recently, medical technologies have moved from a traditional ’one-size-fits-most’ model towards the customisation of mass medicine. Therefore, PDT could be used for customised short-term and long-term treatment by customising medications for individuals based on their current vital organs status, anatomy, unique genetic makeup, behaviour, daily routines, etc. Furthermore, the proposed PDT reference framework could help the next step in personalised medicine by linking the extracted insights and inferences of patients’ organs. For example, wearable sensors and tiny devices such as the BioSticker will be used to collect real-time data and then feed the PDT for the patient, which is connected to their general participator. The general participator will notify the patient of the tests/procedures and the personalised medicine for early prevention [63].**Rapid diagnosis applications:** The proposed PDT reference framework could help early diagnosis by analysing the PDT-based data in addition to genetic information and body measurements to improve the diagnosis of detected and previously unidentified maladies.**Self-care applications:** The proposed PDT reference framework could improve human life by helping self-care application, self-reflection, and personal growth.**Remote-care applications:** The proposed PDT reference framework could help promote remote care procedures for smart healthcare systems by allowing personalised care and reducing the demand for hospitalised services (e.g., reservation, queues, hospital visits, and hospital stay).**Fitness tracker and well-being applications:** The proposed PDT reference framework could help improve fitness tracker and well-being applications for those practising self-care and daily health activities. The PDT can facilitate fitness tracker applications by feeding real-time healthcare data such as heart rate, blood pressure, insulin, and step count.**Medical alert applications:** The proposed PDT reference framework could help understand individualised risk factors. In particular, the system could predict the potential risks by incorporating the individual’s historical data from their medical record aligned with the real-time reads received from live PDT. Based on these predictions, the system will send medical alerts to the corresponding receiver (e.g., person, family member, home care, nurse, doctor, hospital, emergency department, or healthcare provider) to prepare appropriate actions based on the patient’s health conditions.**Pandemic combating** Considering COVID-19 as a pandemic example and with a certain level of privacy, the proposed PDT reference framework could help detect the potential risks to protect people’s lives. First, the updated status within a PDT of a person’s symptoms is analysed. Then, the predicted result is sent to the corresponding receiver (e.g., person, family member, home care, nurse, doctor, hospital, or emergency department) if the informed case is detected. Furthermore, it will be beneficial to report that the COVID-19 infected case situation and notify all people around to practise social distancing and avoid touching and interaction. Consequently, a couple of on-time remote alerts to inform all people around persons infected or potentially infected with COVID-19 can significantly limit pandemic outbreaks [26].

##### Summary

In this section, we provide the high-level requirements identified for building a smart personalised healthcare system by listing the 12 criteria described (see Table 5). Then, we introduce a detailed description of the layers used for building the reference framework of PDT (i.e., physical devices, industrial technologies, and application areas, as can be seen in Figure 8).

## 7. Focusing on Personalised Healthcare Use Cases

Many PDTs span multiple use cases and even categories; these cross-domain personalised healthcare-based use cases form a significant strength of PDTs. Figure 12 depicts the selected use cases: the mitigation of COVID-19 contagion, COVID-19 survivors follow-up care, personalised COVID-19 medicine, personalised osteoporosis prevention, personalised cancer survivor follow-up care, and personalised nutrition. The mitigation of COVID-19 contagion case will be discussed in detail to address the importance of using PDT to protect against the COVID-19 outbreak.

### 7.1. Mitigation of COVID-19 Contagion


**RQ6: How is the PDT concept being applied to protect against the COVID-19 outbreak and any future pandemic?**


In this use case, the PDT is used to mitigate COVID-19 contagion by focusing on the collection, analysis, modelling, and reporting of outbreak data based on the PDTs of people [26]. The PDT-based smart personalised healthcare framework could be used to report people with COVID-19. Then, people around the person with COVID-19 would be notified by their PDTs to start practising social distancing, stop touching shared items, and use hand sanitizer.

We reproduced Figure 13 from our previous work [26] to showcase how the PDTs can be used for mitigating COVID-19 contagion. Based on Figure 13, the participants are people (e.g., potentially infected persons, infected persons, doctors, nurses, or pharmacists), government organisations (e.g., hospitals or health organisations) who communicate and share the updated status of COVID-19 to the blockchain network. People in quarantine areas have permission to share their status with the blockchain networks using their PDTs, as can be seen in the purple arrows in Figure 13. The report from the blockchain network is sent back to the participant, as shown by the blue arrows. Doctors can access the status and suggest the required medicine to the smart contract. Then, the feedback about the medicine from the doctors and pharmacists is sent to the blockchain network—this can be seen in the black arrows in Figure 13.

The COVID-19 pandemic alert can also count the number of cases and identify the areas in which COVID-19 is increasing to mitigate COVID-19 contagion. Consequently, a few on-time remote alerts to notify all people around the infected or potentially infected persons can significantly limit COVID-19 outbreaks. According to Figure 13, the orange and red arrows denote warning and alert, respectively. If the PDT shows that a person is infected with COVID-19, a couple of notifications will be distributed to everyone connected to the network. People around the infected COVID-19 person would be notified by their PDT to start practising social distancing, stop touching shared items, and use hand sanitizer. Furthermore, the PDTs of the people everywhere will analyse their status locally and send up-to-date information on the confirmed positive cases to the pandemic alert.

For example, based on PDTs, as shown in Figure 13, Group 3 will receive a red warning alert due to the confirmation of an infection case which is a neighbour. This alert specifically warns the PDTs of elderly adults because they are likely the age group at high risk. An orange warning alert will be received by the PDTs of Group 2 to warn people to maintain social distancing. This warning alert is sent to analyse their movement and location, as their PDTs send their GPS and proximity sensors. Finally, Group 1 do not receive any alert because there is no confirmed case in proximity to them, and they are practising social distancing. Therefore, the government becomes able to do what is needed to keep the COVID-19 pandemic under control and mitigate COVID-19 contagion, such as by declaring a strict lockdown of the area where the number of positive cases is significantly increasing. Furthermore, as the results of the dynamic updating of healthcare institutions based on the continuous communications to PDTs, the government can relax the restrictions where there is a lower risk and enforce more substantial restrictions where risk is higher.

### 7.2. COVID-19 Survivor Follow-Up Care

After the world started controlling the pandemic outbreak, the healthcare sectors attempted to follow up on COVID-19 survivors (i.e., people infected by COVID-19) and advise that they to adapt to a new lifestyle. Unfortunately, there are not yet studies that confirm the higher-quality evidence for the long-term symptoms of COVID-19 survivors. The most commonly reported symptoms are fatigue, dyspnoea, breathlessness and cough, joint and muscle pain, chest pain, and palpitations. Therefore, healthcare sectors and industries face the challenges of integrating up-to-date medical records for COVID-19 survivors with various telemedicine platforms, data sharing complexities, cost factors, etc. The proposed PDT-based smart personalised healthcare framework could be used to report on the conditions of COVID-19 survivors. The healthcare authorities and providers would have to ask the patients (i.e., COVID-19 survivors) to consent to sharing their data with a large-scale international follow-up study. These data will be collected from their PDTs and then integrated into single personal records for individual COVID-19 survivors. The blockchain network will share the data, and continuous data analysis will be performed. Then, the feedback results will be flagged to the participants and healthcare providers for ongoing patient-specific care for the COVID-19 survivors.

### 7.3. Personalised COVID-19 Medicine

Despite the suffering that the COVID-19 pandemic has caused around the world, there have been benefits. For example, significant improvements have been made in COVID-19 detection and testing. Furthermore, a more accurate diagnosis and COVID-19 personalised medicine will be tailored to each individual. The pandemic has pushed healthcare authorities, industries, and providers towards smart personalised healthcare for individuals [64]. Therefore, scientists and researchers have started collaborating to study how to utilise emerging technologies such as big data, DTs, and AI technologies to create smart personalised healthcare solutions based on individual variables (e.g., immunity system, genomic, and clinical records). Consequently, the proposed PDT-based smart personalised healthcare framework could be used to add valuable progress to this ongoing research. The proposed framework can bring together advanced technologies to build personalised healthcare systems by providing PDT to provide up-to-date health conditions of individuals and people with COVID-19 with appropriate personalised medicine recommendations.

### 7.4. Personalised Osteoporosis Prevention

Knee disorders are pervasive in patients of all ages, ranging from congenital disorders, trauma and sports injuries to osteoarthritis. Furthermore, knee osteoarthritis is the most common disease that causes disability. As the human knee is a complex organ in terms of structure and function, PDTs could monitor human knee health. Then, the PDT-based knee data would allow physicians and physiotherapists to understand the patient’s knee condition better and deliver appropriate personalised care. Furthermore, the PDT-based proposed framework empowers shared decision making between the patient and the treating physician for rehabilitation and value-based health knee care. In particular, the proposed framework combines advanced technologies (i.e., IoT, DTs, AI, and blockchain) to deliver smart personalised solutions for osteoporosis prevention.

### 7.5. Personalised Cancer Survivor Follow-Up Care

All cancer survivors should have follow-up care, which means seeing a healthcare provider for regular medical check-ups once they are finished with their treatment [65]. This follow-up care plan (i.e., survivorship care plan) contains different tests that look for changes in cancer survivor health or any problems due to cancer treatment. The PDT could collect the biomarkers about the cancer survivor such as blood tests, heart rates, steps, sleep, organ data, and other diagnostic data. The proposed PDT-based smart personalised healthcare framework could be used to analyse these biomarkers alongside the cancer historical EMR to build a health curve for that person and compare it with other healthy people. Based on the outcome of the analysis, the doctors can decide whether cancer has returned and whether it is treatable. Then, based on the personal health conditions of the cancer patient, the doctors will decide what surgical intervention should be performed. By utilising the capabilities of the proposed framework, the doctors can run patient-specific detailed datasets through AI and simulation software to identify possible treatment options and see which ones might work best for the individual cancer patient. Furthermore, they can recommend the appropriate personalised care and therapy for the individual cancer patient.

### 7.6. Personalised Nutrition

Nutritional and lifestyle changes remain at the core of healthy ageing and disease prevention [15]. PDT could monitor human body health by reporting clinical and phenotypic variables and behavioural aspects (dietary habits, physical activity tracking sleeping patterns, etc.). The proposed PDT-based smart personalised healthcare framework could be used to analyse this accumulation of healthcare data for individuals. The proposed framework could recommend specific dietary and lifestyle strategies for individuals based on their health and daily lifestyle.

#### Summary

In this section, we provided some selected use cases of adopting PDTs in personalised healthcare, including the mitigation of COVID-19 contagion, COVID-19 survivor follow-up care, personalised COVID-109 19 medicine, personalised osteoporosis prevention, personalised cancer survivor follow-up care, and personalised nutrition. Figure 12 depicts the infographic of the selected use cases of using PDT for smart personalised healthcare industry.

## 8. Validation, Open Challenges, and Discussion

This section discusses the validity of our proposed framework concerning the requirements, followed by the discussion of open challenges.

### 8.1. Validation of Fulfilment Requirements for the Proposed Framework

This section validates the proposed framework. For this purpose, we discuss how the high-level requirements are fulfilled from a technological perspective by using industrial technologies and informative concepts. Table 7 shows the overview of mapping the industrial technologies to the identified requirements for the proposed framework. Further details about the high-level requirements are described in Section 6.1 and Table 5.

For the data collection (R1), smartphones, medical IoT technology, and biosensor technologies are used to collect personal data from various healthcare sources. These collected health data are represented in collaborating DTs to provide the image of a human body. Furthermore, these data are frequently updated to inform on the current status of the human body (R2). The timely updating can be captured using DT technology to maintain rapid updates within physical medical devices. Examples of open source frameworks for creating and managing DTs are Eclipse Ditto, iModel.js, and Mago.

Data management technologies are used to manage the PDT data and reflect the physical and cyber world in real time (R3). For data acquisition, IoT protocols are used for connecting devices, allowing communication, and exchanging data in a structured and meaningful way in the physical and cyber worlds. Some examples of these IoT protocols are: Constrained Application Protocol (CoAP), Message Queuing Telemetry Transport (MQTT), Extensible Messaging and Presence Protocol (XMPP), Data-Distribution Service (DDS), and Advanced Message Queuing Protocol (AMQP). For data query, continuous query processing technologies are used to retrieve the streaming data from connected devices and from the active time-series database [13,66]. Some examples of continuous queries are: InfluxDB, PipelineDB, RethinkDB, and Oracle Continuous Query Language (CQL). Finally, different semantic technologies could be used to describe the PDT model for data modelling. For example, model-to-model technology (e.g., OOP, RDF, and OWL) can be used to describe the relations among collaborating twins to build a concert PDT model.

For data analysis (R4), a set of ML and DL techniques are used to provide smart AI-based personalised healthcare solutions. The most popular ML techniques are: decision tree (DT), K-nearest neighbour (KNN), support vector machine (SVM), random forest (RF), and naive Bayes (NB). In contrast, the DL techniques are convolutional neural network (CNN), recurrent neural networks (RNNs), long short-term memory networks (LSTMs), gated recurrent units (GRU). These techniques are trained and evaluated based on the data-driven PDT to provide timely predictions that help the decision-making process. For data explainability (R5), tools and methods are used to help data scientists design an interpretable ML process. Some examples of these methods are the partial dependence plot (PDP), accumulated local effects (ALE), individual conditional expectation (ICE), local interpretable model-agnostic explanations (LIME), and Shapley additive explanations (SHAP). Finally, for data quality (R6), multiple tools are used for the pre-processing, identifying, cleaning, understanding, and correcting flaws in data that lead to better decision making. There are a set of open source data quality and profiling tools, including Talend Open Studio, Quadient DataCleaner, Open Source Data Quality and Profiling, OpenRefine, DataMatch Enterprise, Ataccama, Apache Griffin, and Power MatchMaker [67].

Furthermore, the DT technology was adopted for its dynamic simulation capability to understand what is happening on the physical device and what could happen in the future (R7). The simulation software simulates individual surgeries and treatments under certain conditions to avoid potential risks. For privacy, confidentiality, and authorisation (R8 and R9), the frequent update of the data for the collaborating twins needs to be exchanged in a secure, trust, authenticated, and transparent process. Therefore, blockchain technology, one application form of DLT, is used. More specifically, some advantages motivated us to adopt blockchain in the proposed framework, including (1) maintaining the trust and secure data exchange among peer-to-peer networks [48,68]; (2) allowing traceability across the entire network [69]; (3) provide insightful consensus-based decision-making process [70]; and (4) deliver efficient solutions by utilising the decentralisation feature of blockchain technology [71]. Different open source blockchain and DLT technologies could be used including HeperLedger, Ethereum, Corda, Quorm, and Openchain. For connectivity (R10), reliable network peers are necessary to communicate efficiently among parties. Different communication technologies could be used such as Beyond Fifth Generation (B5G) and Sixth Generation (6G).

For the decision-making process (R11), ML techniques provide predictions that deliver better insights into a robust and effective decision-making process. Furthermore, the consensus algorithms are used to improve the participants’ collaboration by utilising the agreement of most nodes regarding the potential risk to notify the decision makers within the personalised healthcare systems. Some examples of the use of the consensus algorithms include Proof of Work (PoW), Practical Byzantine Fault Tolerance (PBFT), Proof of Stake (PoS), Proof of Burn (PoB), Proof of Capacity, and Proof of Elapsed Time.

For the computing paradigm (R12), edge-based AI addresses ML algorithms’ processing and implementation locally on the hardware. However, the cloud-based AI performs ML in remote hardware by providing remotely powerful computational resources. Furthermore, the designers and developers are left to decide the best computing paradigm to build a personalised healthcare system (i.e., whether the cloud or the edge is best). Some examples of the open cloud are Apache CloudStack, Eucalyptus, and OpenStack, whereas examples of non-open cloud are Amazon EC2 and Google cloud.

As the PDT can become more person-centric, all DTs have one thing in common: enabling continuous improvement. Therefore, the use case should be identified to successfully implement the PDT solution with a clear understanding of its requirements. Different frameworks are available for building DT/PDTs, such as Eclipse Ditto, Swim OS, and iModel JS. Furthermore, various public cloud vendors are offering DTs that can be used to build PDTs (e.g., Azure Digital Twins, AWS Digital Twins, and IBM Digital Twins). Furthermore, various industrial vendors offering DTs can be used to build PDTs (e.g., GE Predix, Bosch’s digital twin solution, and Siemens MindSphere platform). For the reset modules within the reference framework, Table 7 shows the mapping of the industrial technologies to the identified requirements based on the selected use case.

### 8.2. Open Challenges and Discussion


**RQ7: What are the open challenges to applying PDT in smart personalised healthcare?**


Right now, the PDT is still a vision for the future of the smart personalised healthcare industry. However, there are challenges that must be overcome in order to develop a concrete PDT-based smart personalised healthcare system. Therefore, we will explore and address some of those challenges.

#### 8.2.1. Data Privacy and Regulations

Privacy is a crucial requirement for adopting PDTs to collect personal health data about patients. These data from PDTs are used to build patient-centric models while maintaining patient privacy and data integrity. Therefore, creating a PDT-based smart personalised healthcare system needs to apply strict compliance with data privacy law while allowing the healthcare industry to enhance their deliverable services [19].

#### 8.2.2. Security

Data security is crucial for healthcare data due to hacker attacks, for example, the cyberattack for the Health Service Executive (HSE) of Ireland, which occurred on 14 May 2021 (https://www.hse.ie/eng/services/news/media/pressrel/hse-publishes-independent-report-on-conti-cyber-attack.html, accessed on 21 February 2022). Therefore, it is necessary to ensure the protection of privacy, which is becoming more and more difficult with the increasing functionality of technologies. Like PDTs, patients must be confident that their PDT data are secure, transparent, and accessible. Blockchain technology can be applied to the personalised healthcare industry to protect the access for the PDTs of the patients. Significantly, some initiatives are adopting blockchain technology in medical care/healthcare companies to secure their products and services (see Table 6).

#### 8.2.3. Scalability

Building a DT of an entire system or process plant is a big challenge due to the complex interactions between physical entities. Furthermore, according to the PDT context, PDTs need updating every time physical or operational changes are made to the corresponding person. Therefore, the designers/developers must invest a lot of time and resources to create a customised PDT solution for several participants. Consequently, a scalable solution is needed to handle a large number of PDTs. However, designing a scale PDT-based system is a challenging task which has become one of the most problematic concerns for the personalised healthcare industry.

#### 8.2.4. Data Quality

Data insufficiencies are due to faulty sensors. Furthermore, the biomedical sensors’ data quality plays a vital role in the healthcare system as they are rendered useless if the data quality is bad. Multiple reasons cause bad data quality (e.g., data loss, corruption, weather conditions, and dead sensor battery life) and lead to inconsistent readings. Although the factors that cause insufficient data quality are known, the strategies to overcome data quality problems are still challenges for the healthcare industry.

#### 8.2.5. Modelling

PDT modelling links different engineering features to efficiently handle the personalised healthcare system’s complexity. In particular, the PDT for a human is complex due to the complexity of the human body parts [12]. Furthermore, the communication between human body parts (i.e., collaborating twins) increases complexity. Moreover, the data-driven model methodologies for PDTs should consider the historical patient to feed the analytical process [72]. Thus, building a comprehensive PDTs model is quite challenging to deliver a concrete personalised healthcare system.

#### 8.2.6. Connectivity

With the proliferation of smart devices and DTs, the connection remains a barrier for these devices to achieve their objectives in real time. Because of the critical role of biomedical sensors that built up the PDT, they require high connectivity. If any PDT becomes disconnected, the risk of a delayed healthcare service increases, affecting patients with critical health situations. Therefore, sophisticated communication technologies, such as B5G or 6G, are required.

#### 8.2.7. Timing, Speed and Response

Timing and speed are difficult with regard to performing PDT functions. Firstly, time enhances decision making and reaction times for customer service demands (e.g., potential health risk, appropriate recommendation, personalised treatment planning, rapid disease diagnosis, prevention procedures, and programs), requiring high accuracy and quick replies. In addition, hospitals, healthcare industries, and providers do not want data but want real-time virtual visibility for patients/clients.

#### 8.2.8. Ethics Issues

PDTs for human health face ethical overheads due to the fact that they deal with critical information about people [73,74]. The essential ethical issue is predicting the progressive disease risk and informing and communicating with the patients. Furthermore, the ethical issue is associated with the duration of data access consent. Therefore, all ethical conditions should be specified when dealing with PDTs for human health.

## 9. Conclusions

In this work, we introduced the PDT concept as an enhanced version of the DT with actionable insight capabilities. The PDT helps personalised diagnosis, therapy selection, and procedure planning based on the patient’s physical characteristics, medical history, current condition, and future needs. Furthermore, this can help healthcare providers and the industry who would have personalised information about their clients (i.e., patients) to make timely decisions and personalised recommendations. Then, we proposed a reference framework as a step towards smart personalised healthcare industry. This aims to integrate DTs, blockchain, and AI technologies to deliver smart personalised healthcare services for improving people’s lives. Furthermore, we described the selected personalised healthcare use cases, including the mitigation of COVID-19 contagion, COVID-19 survivor follow-up care, personalised COVID-19 medicine, personalised osteoporosis prevention, personalised cancer survivor follow-up care, and personalised nutrition.

## Figures and Tables

**Figure 1 sensors-22-05918-f001:**
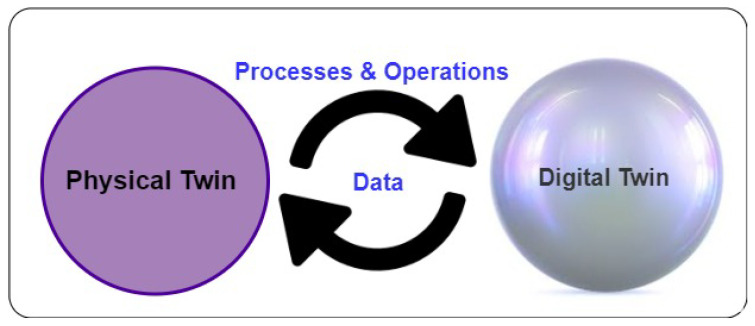
Data synchronisation between the physical twin and digital twin.

**Figure 2 sensors-22-05918-f002:**
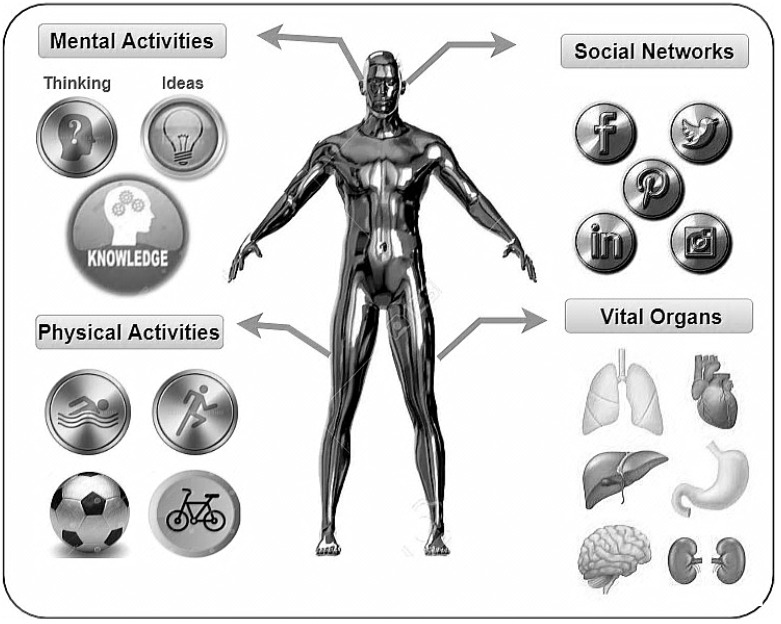
Our comprehensive personal digital twin definition including mental activities, physical activities, social networks, and vital organs.

**Figure 3 sensors-22-05918-f003:**
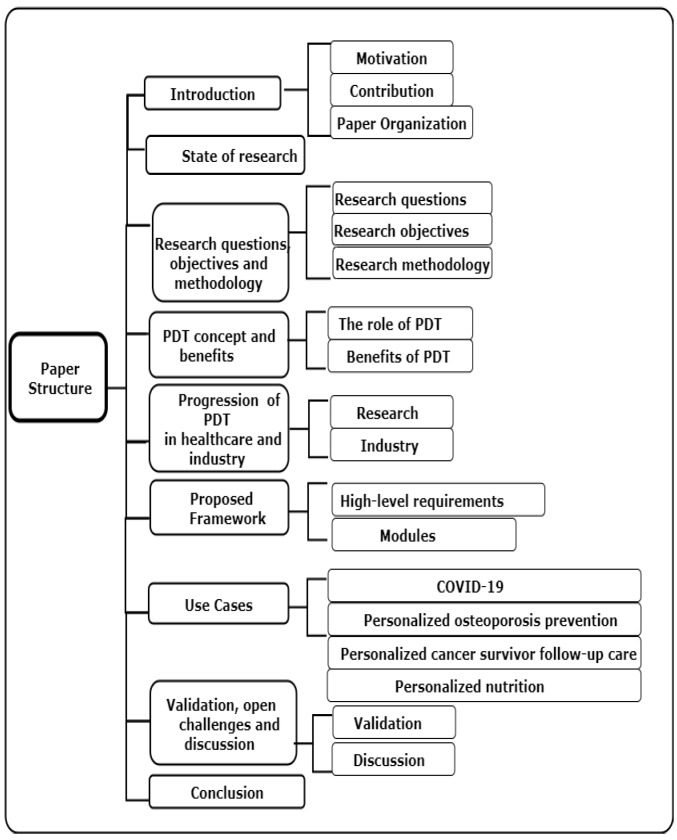
Paper structure.

**Figure 4 sensors-22-05918-f004:**
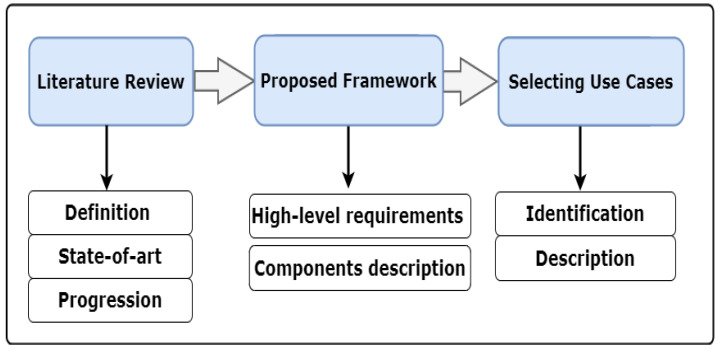
Overview of the research methodology.

**Figure 5 sensors-22-05918-f005:**
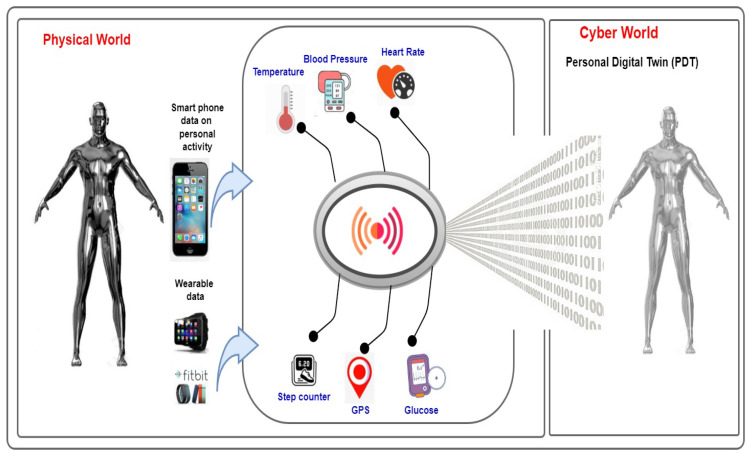
The high level of personal digital twin from a personalised healthcare perspective.

**Figure 6 sensors-22-05918-f006:**
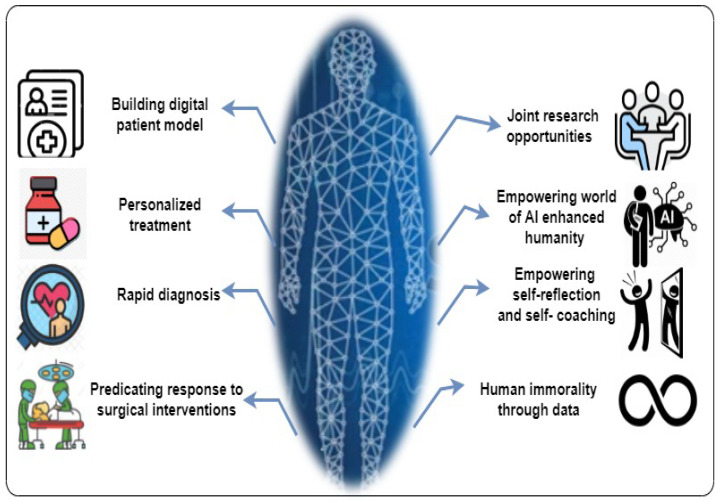
The benefits of using a personal digital twin.

**Figure 7 sensors-22-05918-f007:**
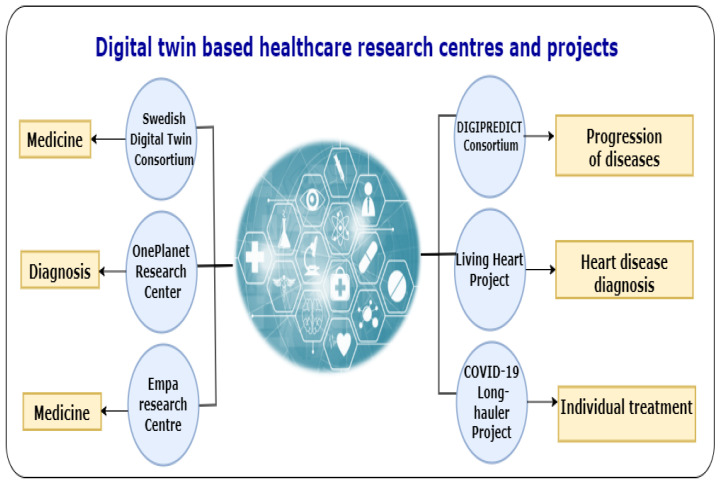
Digital twin-based healthcare research centres and projects and their focus.

**Figure 8 sensors-22-05918-f008:**
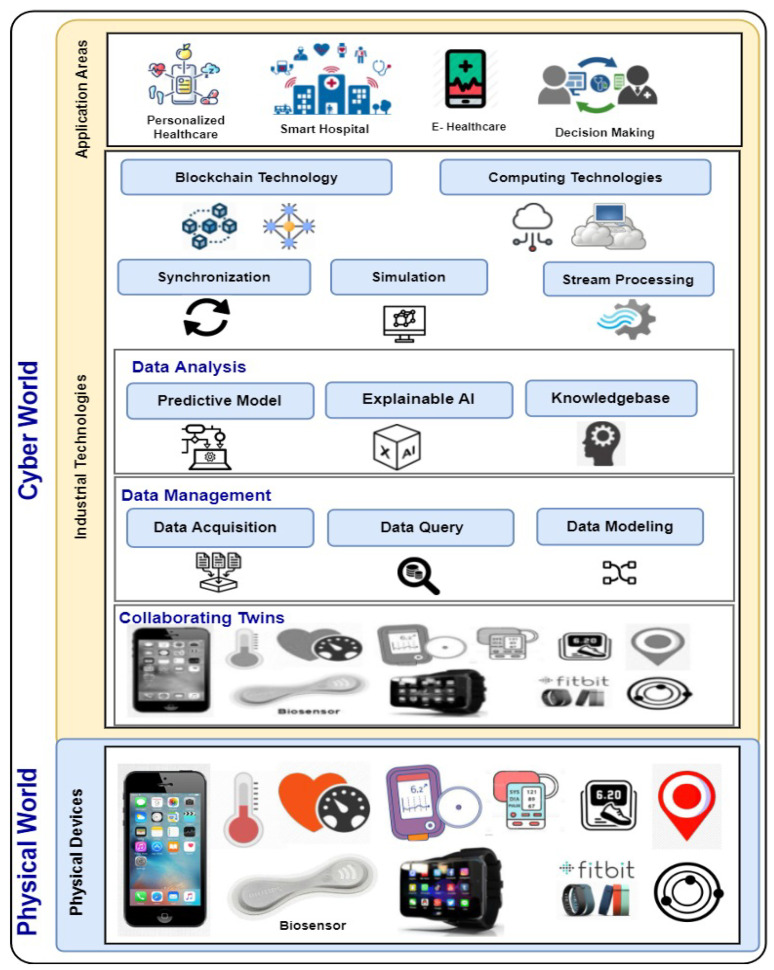
The reference framework of building PDT-based personalised healthcare systems.

**Figure 9 sensors-22-05918-f009:**
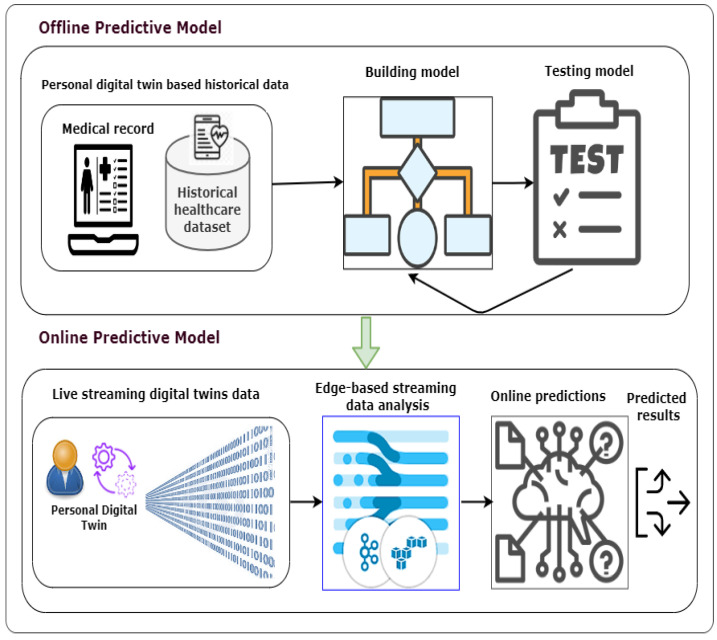
The workflow of building a predictive model based on a personal digital twin.

**Figure 10 sensors-22-05918-f010:**
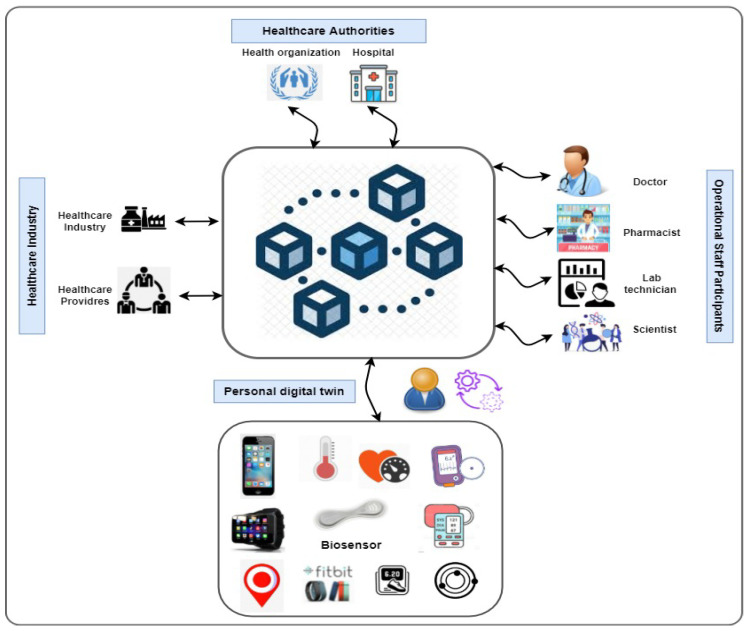
The participants of the blockchain network include personal digital twins, healthcare authorities, healthcare industry, and operational staff participants.

**Figure 11 sensors-22-05918-f011:**
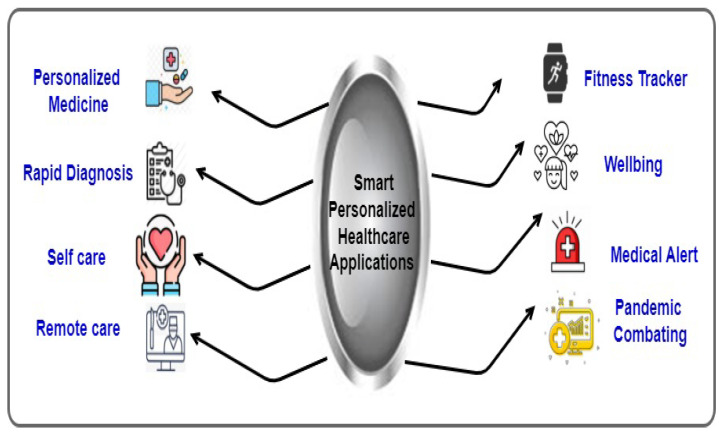
Personal digital twin-based smart personalised healthcare applications areas.

**Figure 12 sensors-22-05918-f012:**
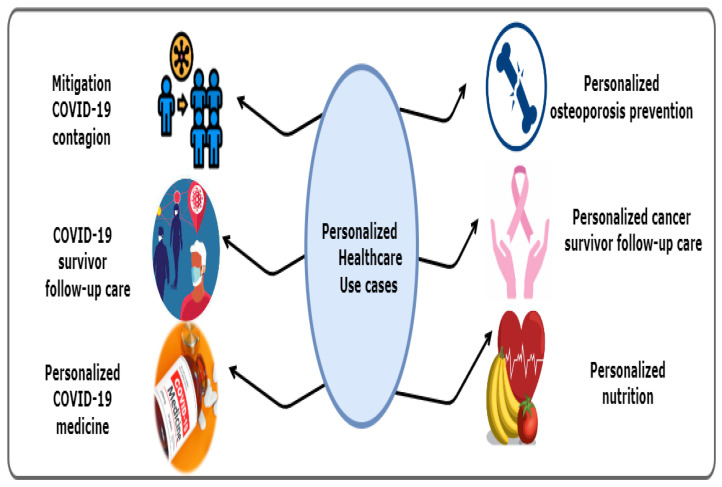
The use cases of using a personal digital twin for a smart personalised healthcare industry.

**Figure 13 sensors-22-05918-f013:**
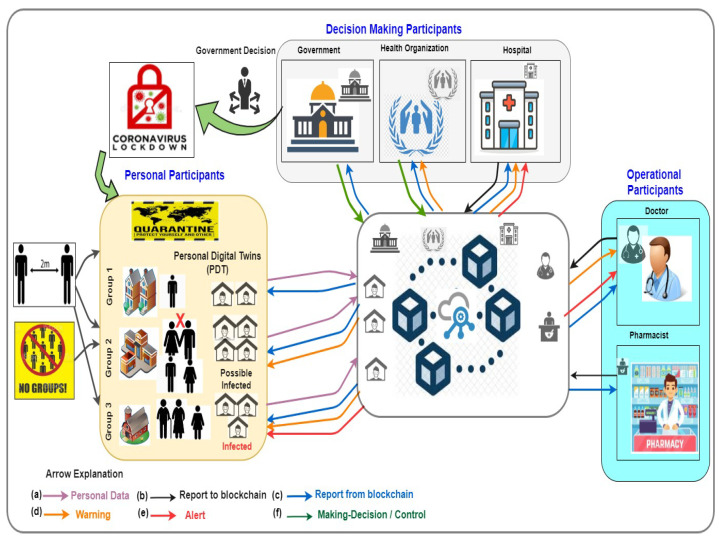
PDT collaboration for mitigating COVID-19 contagion. Data are exchanged among the blockchain-based digital twins network. Arrow explanation: (a) purple arrow is for sending personal data; (b) black arrow is for sending reports to the blockchain network; (c) blue arrow is for receiving reports from the blockchain network; (d) orange arrow is for sending warnings to the cases of infection and potential infection and for sending warnings of the increase in cases to doctors, hospitals, and health organisations; (e) red arrow is for sending alerts to infected cases and for sending warnings of the increase in cases to doctors, hospitals, and health organisation; and (f) green arrow is for sending and broadcasting the decision (e.g., quarantine or lockdown) made by health organisations and governments to the blockchain network (reproduced from [26]).

**Table 1 sensors-22-05918-t001:** Summary of previous research works in personalised healthcare and their limitations.

Reference	Summary	Limitations
[16] (2021)	- Introduces a general-purpose proposal for the creation of DTs. - Introduces DTMS to monitor long-term multiple sclerosis disease	Does not identify high-level requirements for personalised healthcare
[20] (2020)	Introduces the DT concept for personalised medicine	Does not identify mechanisms underlying personalised medicine
[21] (2019)	Presents a vision for applying the DT concept in personalised medical treatment	Limited validity of work
[22] (2021)	Introduces a patient-specific finite element model approach based on DTs for trauma surgery	Does not discuss how existing advanced technologies such AI could help optimise personalised clinical decision making
[25] (2020)	Presents a vision about agent-based DT in the healthcare context	Does not discuss how existing advanced technologies such as AI and blockchain provide more intelligence to DT
[10] (2021)	Discusses how medical DTs are beneficial for protect against viral infection for COVID-19 and any future pandemic	Does not identify high-level requirements to build PDT
[26] (2022)	Introduces a blockchain-based collaborative DTs framework for decentralised epidemic alerting to protect against COVID-19 and any future pandemics	Does not identify high-level requirements for personalised healthcare

**Table 2 sensors-22-05918-t002:** Comparison of the previous works and our current proposed work with respect to the technologies used.

Reference	Highlighted	Digital Twins	Blockchain	Data Analysis/AI and XAI	Applications/Usecases
[16] (2020)	Provides a DT-based general-purpose proposal for healthcare	✓	X	✓	General-purpose proposal
[24] (2021)	Proposes DT-based framework to improve self-management of ergonomic risks for construction work	✓	X	✓	Self-management for construction workers
[22] (2021)	Proposes a patient-specific finite element model approach based on DTs to help personalise clinical decision making	✓	X	X	Personalised clinical decision making
[19] (2021)	Provides a narrative review of existing and future opportunities to capture clinical digital biomarkers in the care of people with multiple sclerosis disease	✓	X	✓	Personalised treatment
[23] (2019)	Proposes a DT-based approach to improve healthcare decision support systems	✓	X	✓	Personalised diagnosis
[21] (2019)	Presents the vision for applying the DT concept in personalised medical treatments	✓	X	✓	Personalised treatment
[17] (2021)	Proposes the conceptual model and characteristics of HDT	✓	X	✓	General-purpose proposal
[18] (2021)	Presents the concept of WDT and its architecture and impact.	✓	X	✓	General-purpose proposal
[25] (2020)	Presents the vision about agent-based DT in healthcare context	X	X	X	Management of traumas
[26] (2022)	Introducing a blockchain-based collaborative DTs framework for decentralised epidemic alerting to protect against COVID-19	✓	✓	✓	Decentralised epidemic alerting
[20] (2020)	Introduces the DT concept for personalised medicine and the steps of the SDTC strategy	✓	✓	X	Personalised medicine

**Table 3 sensors-22-05918-t003:** The research questions and the corresponding section/subsection for the answer.

Number	Description	Section
RQ1	What are the role and benefits of introducing PDT?	Section 4
RQ2	How could PDT revolutionise the personalised healthcare industry?	Section 5
RQ3	What are the requirements for building a PDT-based system for a smart personalised healthcare industry?	Section 6.1
RQ4	What are the key layers for implementing a PDT-based smart personalised healthcare system?	Section 6.2
RQ5	What are the potential applications of using PDT for a smart personalised healthcare industry?	Section 6.2.3
RQ6	How is the PDT concept being applied to protect against the COVID-19 outbreak and any future pandemic?	Section 7.1
RQ7	What are the open challenges to applying PDT in smart personalised healthcare?	Section 8.2

**Table 4 sensors-22-05918-t004:** The progression in the industry for digital twins in relation to smart personalised healthcare.

Company	Description of Product/Service	Type of Product/Service
FEops [27]	A transformation of cardiac images into DTs to improve and expand personalised treatment for patients with structural heart disease	Virtual heart/personalised treatment
Living Brain [30]	Provide a tracking progression of neurodegenerative diseases	Virtual brain
Siemens Healthineers [11]	Provide 3D Digital Heart Twin which is used to simulate surgical procedures and verify tests on patients causing severe injury	Virtual heart
IBM [31]	Efficient and personalised patient treatment using DT model of patient	Personalised treatment
Philips [28]	Using DTs and 3D ultrasound to simulate a virtual heart for providing a heart model and dynamic heart model	Virtual heart
Babylon [32]	Capturing health data from fitness devices and wearables and then transforming them into DTs. The DT-based data are used to support interactions between GPS, doctors, and patients	Personalised healthcare
DigiTwin [33]	Converting 2D patient medical images (MRI, CT scans) into 3D virtual images to allow clinicians to engage patients with their DTs for improving patient education and shared decision-making processes leading to better treatment plans	Personalised treatment
Dassault Systèmes [29]	Provide 3D models of live hearts which are used for cardiac research purpose	Virtual heart

**Table 5 sensors-22-05918-t005:** High-level requirements for a smart personalised healthcare system based on PDT.

Req. No.	Requirement	Reason
R1	Data collection	supporting data-driven smart personalised healthcare
R2	Data update frequency	providing real-time update on the physical twin
R3	Data management	Maintaining data management including data acquisition, data query, and data modelling
R4	Data analysis	Enabling advanced predictions of the potential risks, customised medicine, treatment planning, etc.
R5	Data explainability	Supporting clinical decision systems
R6	Data quality	Leading to better decision making
R7	Simulation capabilities	Enabling virtual visibility
R8	Privacy and confidentiality	Maintaining the confidentiality of the patient’s personal information including their medical records
R9	Authorisation	Allowing the authorised people by law to access the people personal information
R10	Connectivity	Allowing to connect the on-body sensors and wearable sensors to their digital twins
R11	Decision making	Providing an insightful decision-making process
R12	Computing paradigm	Performing analysis (e.g., cloud and edge)

**Table 6 sensors-22-05918-t006:** Blockchain in the healthcare industry.

Company	Industry	Location	Description	Blockchain Application Usage	Real-Life Impact in Healthcare
Akiri [51]	Big data	Foster City, CA	Providing patient health data protection using ledger technology	Using ledger technology	Security, sharing authorisation
BurstIQ [52]	Big data, cybersecurity	Colorado Springs, Colorado	Helping healthcare companies secure patient data	Improve medical data sharing	Prescription drugs
Factom [53]	IT, enterprise software	Austin, Texas	Creating a product to help the healthcare industry securely store digital health records	Securely store digital health records	Data security
MEDICALCHAIN [54]	Electronic health records, medical	London, England	Maintaining the integrity of health records	Maintain patients’ records and protect patient identity	Consultations
Guardtime [55]	Cybersecurity, blockchain	Irvine, California	Helping healthcare sectors implement blockchain into their cybersecurity methods	Apply for blockchain for cybersecurity in healthcare	Deploying blockchain platforms
Professional Credentials Exchange [56]	Big data	Tampa, FL	Creating a distributed ledger of healthcare credentials data	Fulfil the requirements of data sharing and authorisation	Verify the credentials of patient’s data
Coral Health [58]	Healthcare, IT	Vancouver, Canada	Providing automated healthcare services by using ledger technology	Use ledger technology to connect parties and smart contract between patients and doctors	Tracking patients
Robomed [59]	Blockchain, medicine	Moscow, Russia	Offering patients a single point of care using AI and blockchain	Use blockchain to gather patients’ information and share it with patients’ healthcare providers	Security and sharing medical data
Patientory [60]	Blockchain, cybersecurity, healthcare, IT	Atlanta, Georgia	Provide blockchain-based platform to help the healthcare industry securely transfer their information via blockchain	Enabling the secure storage and transfer of important medical information.	Security and data storage

**Table 7 sensors-22-05918-t007:** Validation of fulfilment requirements from the technology perspective for the proposed framework.

Req. No.	Main Requirements	Enabled by Industrial Technologies	Examples
R1	Data collection	Smartphones and medical IoT technology	Biosensors and wearable devices
R2	Data update frequency	DT technology	Eclipse Ditto, iModel.js, Mago3d
R3	Data management	For data acquisition: IoT protocols	CoAP, MQTT, XMPP, DDS, AMQP
		For data query: continuous query processing	InfluxDB, PipelineDB, RethinkDB
		For data modelling: semantic technology	OOP, RDF, OWL
R4	Data analysis	Machine learning techniques	DT, KNN, SVM, RF, NB
		Deep learning techniques	CNN, RNN, LSTM, GRU
R5	Data explainability	Interpretable methods for machine learning	PDP, ALE, ICE, LIME, SHAP
R6	Data quality	Open source data quality and profiling tools	Talend Open Studio, Quadient DataCleaner, OpenRefine, DataMatch Enterprise, Ataccama, Apache Griffin, Power MatchMaker
R7	Simulation capabilities	DT technology	Ditto, Swim OS, iModel.js
R8	Privacy and confidentiality	Blockchain and DLT technology	HeperLedger, Ethereum, Corda, Quorm, Openchain
R9	Authorisation	Blockchain and DLT technology	HeperLedger, Ethereum, Corda, Quorm, Openchain
R10	Connectivity	Wireless communication technologies	Beyond Fifth Generation (B5G) Sixth Generation (6G), WiFi
R11	Decision making	Machine learning techniques	DT, KNN, SVM, RF, NB
		Consensus algorithms	PoW, PBFT, PoS, PoB
R12	Computing paradigm	Cloud, edge, etc.	Open cloud: Apache CloudStack, Eucalyptus, OpenStack Not open cloud: Amazon EC2, Google cloud

## Data Availability

No Data available online. For further query email to corresponding author (rsahal@ucc.ie).

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
