# Peer review of "Personal Digital Twin: A Close Look into the Present and a Step towards the Future of Personalised Healthcare Industry"

_sensors, 2022, doi:10.3390/s22155918_

Round 1

Reviewer 1 Report

The authors presented a survey on the concept of Personal Digital Twin (PDT) as an enhanced version of the DT with actionable insights capabilities. The idea is interesting. However, I have the following concerns. 

1. Please revise the grammatical errors of the paper. 

2. Recent research containing emerging technologies such as IoT, AI, and Blockchain is missing. Some are mentioned as follows. 

-> "FBI: A Federated Learning-Based Blockchain-Embedded Data Accumulation Scheme Using Drones for Internet of Things," in IEEE Wireless Communications Letters, vol. 11, no. 5, pp. 972-976, May 2022, doi: 10.1109/LWC.2022.3151873.

-> A Blockchain Based System for Healthcare Digital Twin," in IEEE Access, vol. 10, pp. 50523-50547, 2022, doi: 10.1109/ACCESS.2022.3173617.

3. Please improve the quality of Fig. 3, 4.

4. Please highlight novelty. 

5. A table is required for Section 2 containing a summary and limitations of the current works. 

6. Add a summary containing the discussion at the end of each section to improve readability. 

7. A discussion on scalability is also required. 

8. Data in personal digital twin in healthcare are sensitive data. A discussion on privacy is also required. 

9. A hint on the implementation of PDT in the real world would be better to increase the feasibility of the proposed scheme.

Reviewer 2 Report

I want to express the interest in your work and your vision of future healtcare and underline important your paper. The transition from the unnamed set of numbers to a personal 3d model of a patien may be a great step forward in this area. 

Despite the fact the work has a very big potential and shows us the ways to solve the problem of development of the future of our healthcare, need to emphasize your view on few issues relataed to formatting of the work, what in fact improves the overall perception of the results presented in the article and ease of reading. 

For your pleasure I split up the peer review into issues and their solutions (if applicable). Hope my advices may be helpful in your furter the research.

Issue 1. The “cross” sign is used under the Correspondence section but can be omitted. 

Issue 2. Most of the images and tables have a different alignment and size on the page and don’t answer the the next requirement: the images and tables should be placed between left and right border of the page and can’t exceed it (see solutions).

Issue 3. Research questions “RQ” (line 199-238)  form a separate sections of the paper and have the same emphasize and importance as the section titles. They need to follow the indentation rules and use other formatting and presented as bulleted list.

Issue 4. Subsection 6.2.2. has a multiple subsections of its own, but they have the same formatting as a plain text. In this case I suggest using other type of formatting (Bold) and include them within the same paragraph, what is starting after them. Correct example to solve this issue can ne seen on lines 565 and 579.

Issue 5. Line 905. The references and footnotees should be placed within the text they are connected to. This issue can be easily fixed by replacing 14 May 2021 with 14/05/2021 which gives the necessary space.

Issue 6. Line 961. The titles of the sections and subsections of the paper can’t be placed on the last line of the page. It can be solved by placing one whitespace after block of text.

Now after listing the issues, I supply the necessary solutions in order to fix them (it’s up to you to select other acceptable way to solve them).

Solulutions of issue 2. The figures 2, 5, 6, 7, 9, 11, 12, 13 have to be resized down to fill the space between the left and right border of the text (the following figure names should also follow this requirement). Figure 3 has to be modified in the way where blocks “Mitigation COVID-19 contagion”, “COVID-19 survivior follow-up care” and “Personalized COVID-19 medicine” are either omitted, or they should substitute the block “COVID-19”. The issue with table 1 can be done as follows: the column Reference and Blockchain has to squeezed to fit the allowed width. Table 2 has to be aligned to the center, like figure 4 below. The issue with table 3 can be done the same way as of issue with table 1, squeezing the colunm Company. The issue with table 4 can be solved the same way as of previous, squeezing the column Requirement. Issue with table 5 can be done by removing the column Location and joining the columns Company and Industry into single column. The issue with table 6 can be solved by removing the column Main requirements and adding the footnone “The detailed requirements are presented in the Table 4”.

Solution of Issue 3. Since the text after each section title have to start from indentation, I suggest using a preceding text before definition of research question (RQ), so as it forms separate task definition for separate section of the text. In general, the Research questions are divided into separate section of the article (Task definition) and then the following text is solving them one after another. In your case, it will be a good idea to left the definition of the task/problem as a question: e.g: “In previous sections we discussed the importance of the visual analysis of the health paramenters, but after recent advances in computer tomography, 3d xray imaging, one may ask the question: “what makes them so special and why the medical specialists prefer the raw imaging, often in different views over written text, made after diagnosing the problems in health of their patients (Research question 0)?””

Reviewer 3 Report

This paper investigated the concept and progression of personal digital twin in healthcare research and industry, and presented a reference framework of PDT for smart personalized healthcare. The topic is interesting and important considering PDT has the potential to enable more personalized, intelligent and proactive healthcare with the advances of IoT, big data, and AI technologies.

This paper has the potential to be accepted, but some important points have to be clarified or fixed.

1. The digital twin is a virtual replica for a human including mental, biological, physical and social perspectives. Why does the high-level model of PDT in figure 5 and the reference framework in figure 8 only present physical devices which are inadequate to build a digital model for a particular patient? The framework for PDT should provide the demographics, medical history and other information of the physical world for healthcare management and clinical decision making.

Some relevant references:

[1] Wickramasinghe, Nilmini et al. “A Vision for Leveraging the Concept of Digital Twins to Support the Provision of Personalised Cancer Care.” IEEE Internet Computing (2021): 1-1. (Figure 2)

[2] Ricci, A. et al. “Pervasive and Connected Digital Twins – A Vision for Digital Health.” IEEE Internet Computing (2021): 1-1. (Figure 1)

2.  Its unclear to me the synchronization of the physical twin and the digital twin illustrated in Figure 1. Should the physical twin share data for the digital twin, and the digital twin provide knowledge and information to guide the process/operations in the physical world?

References:

Rathore, Muhammad Mazhar Ullah et al. “The Role of AI, Machine Learning, and Big Data in Digital Twinning: A Systematic Literature Review, Challenges, and Opportunities.” IEEE Access 9 (2021): 32030-32052. (Figure 4)

3. Figure 2 can be improved to illustrate in a consistent style.

Round 2

Reviewer 1 Report

I am recommending to accept this paper.